# Stability Analysis of Shear Deformable Inhomogeneous Nanocomposite Cylindrical Shells under Hydrostatic Pressure in Thermal Environment

**DOI:** 10.3390/ma16134887

**Published:** 2023-07-07

**Authors:** Abdullah H. Sofiyev, Nicholas Fantuzzi

**Affiliations:** 1Department of Mathematics, Istanbul Ticaret University, Beyoglu, Istanbul 34445, Türkiye; 2Scientific Research Centers for Composition Materials, UNEC Azerbaijan State Economic University, Baku 1001, Azerbaijan; 3Scientific Research Centers for Composite Constructions, Odlar Yurdu University, Baku 1072, Azerbaijan; 4Department of Civil, Chemical, Environmental, and Materials Engineering, University Bologna, 40136 Bologna, Italy; nicholas.fantuzzi@unibo.it

**Keywords:** nanocomposites, inhomogeneity, cylindrical shell, stability, hydrostatic pressure, buckling pressure

## Abstract

In this study, the stability of inhomogeneous nanocomposite cylindrical shells (INCCSs) under hydrostatic pressure in a thermal environment is presented. The effective material properties of the inhomogeneous nanocomposite cylindrical shell are modeled on the basis of the extended mixture rule. Based on the effective material properties, the fundamental relations and stability equations are derived for thermal environments. In this process, the first-order shear deformation theory (FSDT) for the homogeneous orthotropic shell is generalized to the inhomogeneous shell theory. This is accomplished using the modified Donnell-type shell theory. The analytical expressions are obtained for hydrostatic buckling pressure of INCCSs in the framework of FSDT and classical shell theory (CST) by obtaining a solution based on Galerkin’s procedure. The numerical examples presented include both comparisons and original results. The last section shows the influences of carbon nanotube (CNT) models, volume fraction, and shell characteristics on the hydrostatic buckling pressure in the thermal environment.

## 1. Introduction

Cylindrical shells are used in a wide variety of fields such as aircraft and spacecraft, submarines, rockets, and nuclear reactors. The properties of the composite materials used for the manufacture of cylindrical shells should be at a level that can meet the requirements of modern technology. At the end of the 20th century, the extraordinary efforts of materials scientists led to successful results, and in 1991, a seminal composite material with extraordinary properties, called carbon nanotubes, was discovered [1]. Being extremely small and light, the resources required to manufacture them made this material advantageous, as many can be produced with only a small amount. Initially, despite the extremely low densities of carbon nanotubes, their high tensile strength, high toughness, and wear resistance were demonstrated by experimental and theoretical studies. Then, the extraordinary properties of CNTs enabled them to be used as reinforcement in structural elements made of polymer, ceramic, metal, and other materials and to form nanocomposites [2,3,4].

In recent years, developments in nanotechnology have facilitated the design and production of new-generation advanced functionally graded materials and extended their application areas. Nanocomposites (NCs) are a new class of inhomogeneous materials that are used in the aerospace, aviation, and automotive industries because they allow the production of structural elements that use few raw materials, and they are affordable and lightweight and use less energy. In addition, polymer structural members reinforced with CNTs can exhibit high mechanical properties, high toughness, improved thermal and electrical properties, good optical clarity, high wear resistance, and other outstanding properties, despite the slight increase in weight [5,6,7,8,9].

After research on nanocomposites revealed that they have extraordinary mechanical, physical, and chemical properties [10,11,12], Shen [13] created a new concept of the stability of shells consisting of various patterned nanocomposites and opened up a new field for setting and solving a number of stability and vibration problems. This concept was experimentally confirmed by Kwon et al. [14]. Following this work, Shen and co-authors [15,16] investigated the postbuckling of nanotube-reinforced composite cylindrical shells under axial and radial mechanical loads in thermal environments using a singular perturbation technique. Brischetto and Carrera [17] proposed classical and refined shell models for the analysis of nano-reinforced structures. Aragh [18] presented mathematical modeling of the stability of carbon nanotube-reinforced panels. Garcia-Macias et al. [19] investigated buckling analysis of functionally graded CNT-reinforced curved panels under axial compression and shear. Summaries of information on the studies of inhomogeneous nanocomposite structural elements, especially cylindrical shells, between 2012 and 2020 are included in review articles of Khaniki and Ghayesh [20], Liew et al. [21], and Garg et al. [22]. In the vast majority of studies conducted on this subject in the last three years, different aspects of the stability and instability of unconstrained CNT-patterned cylindrical panels and shells in different environments have been investigated by using different methods within the framework of different shell theories. Among them, Tocci Monaco et al. [23] investigated the hygro-thermal vibrations and buckling of laminated nanoplates via nonlocal strain gradient theory. Bacciocchi [24] examined the buckling analysis of three-phase CNT/polymer/fiber functionally graded orthotropic plates and discussed the influence of the non-uniform distribution of the oriented fibers on the critical load. Hieu and Tung [25] presented the buckling of shear deformable CNT-patterned cylindrical shells and toroidal shell segments under mechanical loads in thermal environments. Tocci Monaco et al. [26] reported the critical temperatures for vibrations and buckling of magneto-electro-elastic nonlocal strain gradient plates. Cornacchia et al. [27] presented an analytical solution for linear vibrations and buckling problems of cross- and angle-ply nanoplates with strain gradient theory. Tocci Monaco et al. [28] presented the trigonometric solution for the bending analysis of magneto-electro-elastic strain gradient nonlocal nanoplates in a hygro-thermal environment. Izadi et al. [29] studied the torsional characteristics of CNTs by using micropolar elasticity models and a molecular dynamics (MD) simulation. Hieu and Tung [30] reported the thermal buckling and postbuckling of a CNT-reinforced composite cylindrical shell surrounded by an elastic medium with tangentially restrained edges. Chakraborty et al. [31] investigated the instability characteristics of damped CNT-reinforced laminated shell panels subjected to in-plane excitations and thermal loading. Khayat et al. [32] investigated the effect of uncertainty sources on the dynamic instability of CNT-reinforced porous cylindrical shells integrated with piezoelectric layers under electro-mechanical loadings. Ghasemi and Soleymani [33] examined the effects of CNT distribution on the buckling of carbon nanotubes/fiber/polymer/metal hybrid laminate cylindrical shells. Avey et al. [34] presented the mathematical modeling and analytical solution of the thermoelastic stability problem of functionally graded nanocomposite cylinders within different theories. Shahmohammadi et al. [35] studied the nonlinear thermo-mechanical static analysis of toroidal shells made of nanocomposite/fiber-reinforced composite plies surrounded by an elastic medium. Sofiyev et al. [36] investigated the buckling behavior of sandwich cylindrical shells covered by functionally graded coatings with clamped boundary conditions under hydrostatic pressure. Sun et al. [37] examined the postbuckling analysis of GPL-reinforced porous cylindrical shells under axial compression and hydrostatic pressure. Trang and Tung [38] investigated the thermoelastic stability of thin CNT-reinforced composite cylindrical panels with elastically restrained edges under non-uniform in-plane temperature distribution. Avey et al. [39] presented the thermoelastic stability of CNT-patterned conical shells under thermal loading in the framework of FSDT. Avey et al. [40] examined the mathematical modeling and solution of the nonlinear vibration problem of laminated plates with CNT-originating layers interacting with a two-parameter elastic foundation. Izadi et al. [41] studied the bending characteristics of CNTs using micropolar elasticity models and MD simulations. Ipek et al. [42] investigated the buckling behavior of nanocomposite plates with functionally graded properties under compressive loads in elastic and thermal environments.

The literature review reveals that the stability of INCCSs subjected to hydrostatic pressure in a thermal environment is not sufficiently investigated. In this work, the FSDT proposed by Ambartsumian [43] for the homogeneous anisotropic shells is generalized to INCCSs. This is one of the original aspects of the study. Unlike some studies mentioned above, the shear stress functions are used instead of the shear correction factor for INCCSs in this study. It should be emphasized that in inhomogeneous nanocomposite structural elements, it is unrealistic to use the fixed-valued shear correction factor, since the value of the shear correction factor will change when the shape of the CNT patterns is changed. The Shapery model is used for the coefficients of thermal expansion, which are functions of the thickness coordinate [44]. The accuracy of the present method for buckling analyses of cylindrical shells subjected to hydrostatic pressure is confirmed by two comparative studies by Kazagi and Sridharan [45] using the finite element method (FEM), and Shen and Noda [46] using high-order shear deformation theory (HSDT) and a singular perturbation technique.

This paper is structured as follows: the description of the model is presented in Section 2, Section 3 includes the derivation of governing equations, the solution procedure is performed in Section 4, and Section 5 includes comparative and specific examples.

## 2. Description of the Model

The notes on the geometry of the inhomogeneous nanocomposite cylindrical shell subjected to hydrostatic pressure are illustrated in Figure 1. The geometrical parameters such as length, radius, and thickness of the INCCS are designated by L, R, and h, respectively (Figure 1a). The shell displacements surface in the $x_{1},\;x_{2}$, and $x_{3}$ directions are designated by u, v, and w, respectively.

The hydrostatic pressure acting on the inhomogeneous nanocomposite cylindrical shell is expressed as follows (Figure 1b) [47]:(1)$$N_{11}^{0}=-P\cdot R/2,\;N_{22}^{0}=-P\cdot R,\;N_{12}^{0}=0$$
where $N_{ij}^{0}$ are the membrane forces for the condition with zero initial moments.

The INCCS is assumed to be formed by the reinforcement of the homogeneous isotropic polymer with single-walled CNTs (SWCNTs). The SWCNT reinforcement is aligned in the $x_{1}$ direction and distributed either uniformly (U) or inhomogeneously (IN) in the thickness direction of the shell.

It is assumed that the material properties of the CNTs and the matrix are temperature-dependent. Therefore, the effective material properties of INCCSs, such as Young’s modulus, shear modulus, and thermal expansion coefficients, are functions of temperature and location. Since the effective Poisson’s ratio and density are weakly dependent on temperature change and location, they are considered constant [13,15]. Considering these assumptions, the micromechanical model of the mechanical and thermal properties of INCCSs can be constructed as follows:(2)$$\begin{array}{l} E_{11}^{\left({\bar{x}}_{3},T\right)}=\eta_{1}V_{cnt}E_{11}^{cnt}(T)+V_{m}E^{m},\;\frac{\eta_{2}}{E_{22}^{\left({\bar{x}}_{3},T\right)}}=\frac{V_{cnt}}{E_{22}^{cnt}(T)}+\frac{V_{m}}{E^{m}(T)},\;\frac{\eta_{3}}{G_{12}^{\left({\bar{x}}_{3},T\right)}}=\frac{V_{cnt}}{G_{12}^{cnt}(T)}+\frac{V_{m}}{G^{m}(T)},\;\\ G_{13}^{\left({\bar{x}}_{3},T\right)}=G_{12}^{\left({\bar{x}}_{3},T\right)},\;G_{23}^{\left({\bar{x}}_{3},T\right)}=1.2G_{12}^{\left({\bar{x}}_{3},T\right)},\;\nu_{12}=V_{cnt}^{*}\nu_{12}^{cnt}+V_{m}\nu^{m},\;\rho_{12}=V_{cnt}^{*}\rho_{12}^{cnt}+V_{m}\rho^{m} \end{array}$$
where the total volume fraction of CNTs is determined from the following expression:(3)$$V_{cnt}^{*}=\frac{m_{cnt}}{m_{cnt}+(\rho^{cnt}/\rho^{m})\left(1-m_{cnt}\right)}\;$$
in which $m_{cnt}$ is the mass fraction of the CNT, and $\rho^{cnt}$ and $\rho^{m}$ are the densities of the CNT and matrix, respectively. $E_{11}^{cnt},\;E_{22}^{cnt},\;G_{12}^{cnt},\;\rho^{cnt}$, and $\nu_{12}^{cnt}$ indicate the Young’s modulus, shear modulus, density, and Poisson ratio of the CNT, respectively, $E^{m},\;G^{m},\;\rho^{m}$, and $\nu^{m}$ indicate similar properties of the polymer. $V_{cnt}$ and $V_{m}$ are the volume fractions of CNT and the matrix, which are related by $V_{cnt}+V_{m}=1$. Moreover, $\eta_{j}(j=1,\;2,\;3)$ are the CNT efficiency parameters defined to account for the size dependence of the resulting nanocomposites. The magnitudes of the CNT efficiency parameter are determined by comparing the modulus of elasticity of the nanocomposites obtained from the molecular dynamics simulation with those estimated from the mixing rule [13].

Since the coefficients of thermal expansion are functions of the thickness coordinate, their mathematical expression in the longitudinal and transverse directions is expressed by the Shapery model as follows [44]:(4)$$\begin{array}{l} \alpha_{11}^{\left({\bar{x}}_{3},T\right)}=\frac{V_{cnt}E_{11}^{cnt}(T)\alpha_{11}^{cnt}(T)+V_{m}E^{m}(T)\alpha^{m}(T)}{V_{cnt}E_{11}^{cnt}(T)+V_{m}E^{m}(T)}\;,\;\\ \alpha_{22}^{\left({\bar{x}}_{3},T\right)}=(1+\nu_{12}^{cnt})V_{cnt}\alpha_{22}^{cnt}(T)+(1+\nu^{m})V_{m}\alpha^{m}(T)-\nu_{12}\alpha_{11}^{\left({\bar{x}}_{3},T\right)} \end{array}$$
where $\alpha_{11}^{cnt},\;\alpha_{22}^{cnt}$, and $\alpha^{m}$ indicate thermal expansion coefficients of the CNT and the matrix. 

It is assumed that the CNT distribution in NCs is linearly graded in the thickness direction since the possibilities of contemporary technologies can provide linear variation of the volume fraction. Three types of patterns, namely Λ-, X-, and V-models, are considered, apart from the uniform (U) distribution. The volume fraction of these patterns is modeled as follows [13]:(5)$$\begin{array}{l} V_{cnt}=\left(1+2{\bar{x}}_{3}\right)V_{cnt}^{*}\;\mathrm{for}\;\Lambda \\ V_{cnt}=4\left|{\bar{x}}_{3}\right|V_{cnt}^{*}\;\mathrm{for}\;X\\ V_{cnt}=\left(1-2{\bar{x}}_{3}\right)V_{cnt}^{*}\;\mathrm{for}\;V \end{array}$$

In the U-model of CNTs along the thickness, one has $V_{cnt}=V_{cnt}^{*}$.

The configurations of the homogeneous U-model and three types of INCs are illustrated in Figure 2. 

## 3. Governing Equations

The stress–strain relationships of inhomogeneous nanocomposite cylindrical shells in thermal environments within FSDT can be expressed as follows [34,39]: (6)$$\left[\begin{array}{l} \tau_{11}\\ \tau_{22}\\ \tau_{12}\\ \tau_{13}\\ \tau_{23} \end{array}\right]=\left[\begin{array}{lllll} a_{11}^{\left({\bar{x}}_{3},T\right)} & a_{12}^{\left({\bar{x}}_{3},T\right)} & 0 & 0 & 0\\ a_{21}^{\left({\bar{x}}_{3},T\right)} & a_{22}^{\left({\bar{x}}_{3},T\right)} & 0 & 0 & 0\\ 0 & 0 & a_{66}^{\left({\bar{x}}_{3},T\right)} & 0 & 0\\ 0 & 0 & 0 & a_{55}^{\left({\bar{x}}_{3},T\right)} & 0\\ 0 & 0 & 0 & 0 & a_{44}^{\left({\bar{x}}_{3},T\right)} \end{array}\right]\left[\begin{array}{l} \varepsilon_{11}\\ \varepsilon_{22}\\ \gamma_{12}\\ \gamma_{13}\\ \gamma_{23} \end{array}\right]+\left[\begin{array}{l} \tau_{1T}\\ \tau_{2T}\\ 0\\ 0\\ 0 \end{array}\right]$$
where $\tau_{iT}(i=1,2)$ are defined as
(7)$$\tau_{iT}=-\frac{E_{ii}^{\left({\bar{x}}_{3},T\right)}\alpha_{ii}^{\left({\bar{x}}_{3},T\right)}T({\bar{x}}_{3})}{1-\nu_{12}\nu_{21}}\;$$
in which $\tau_{ij}(i,j=1,2,3)$; $\varepsilon_{ii}$ and $\gamma_{ij}$ are the stresses and strains of the INCCSs, respectively; and the coefficient $a_{ij}^{\left({\bar{x}}_{3},T\right)}$ denotes the stiffness matrix, and its elements are expressed as
(8)$$\begin{array}{l} a_{11}^{\left({\bar{x}}_{3},T\right)}=\frac{E_{11}^{\left({\bar{x}}_{3},T\right)}}{1-\nu_{12}\nu_{21}},\;a_{22}^{\left({\bar{x}}_{3},T\right)}=\frac{E_{22}^{\left({\bar{x}}_{3},T\right)}}{1-\nu_{12}\nu_{21}},\;a_{12}^{\left({\bar{x}}_{3},T\right)}=\frac{\nu_{21}E_{11}^{\left({\bar{x}}_{3},T\right)})}{1-\nu_{12}\nu_{21}}=\frac{\nu_{12}E_{22}^{\left({\bar{x}}_{3},T\right)}}{1-\nu_{12}\nu_{21}}=a_{21}^{\left({\bar{x}}_{3},T\right)},\;\\ a_{44}^{\left({\bar{x}}_{3},T\right)}=G_{23}^{\left({\bar{x}}_{3},T\right)},\;a_{55}^{\left({\bar{x}}_{3},T\right)}=G_{13}^{\left({\bar{x}}_{3},T\right)},\;a_{66}^{\left({\bar{x}}_{3},T\right)}=G_{12}^{\left({\bar{x}}_{3},T\right)} \end{array}$$

The shear stresses of INCCSs within FSDT change in the thickness direction as follows [34,39,43]:(9)$$\tau_{13}=\frac{df(x_{3})}{dx_{3}}F_{1}(x_{1},x_{2}),\;\tau_{23}=\frac{df(x_{3})}{dx_{3}}F_{2}(x_{1},x_{2})$$
where $F_{i}(x_{1},x_{2}),\;(i=1,2)$ indicate the normal rotations to the mid-surface versus $x_{2}$ and $x_{1}$ axes, respectively, and $f\left(x_{3}\right)$ denotes an a posteriori specified shape function and is defined as $f\left(x_{3}\right)=x_{3}-4x_{3}^{3}/3h^{2}$ [43].

The relations $\varepsilon_{11},\varepsilon_{22}$, and $\gamma_{12}$ for the INCCSs within the FSDT can be expressed as follows:(10)$$\left[\begin{array}{l} \varepsilon_{11}\\ \varepsilon_{22}\\ \gamma_{12} \end{array}\right]=\left[\begin{array}{l} \varepsilon_{11}^{0}\\ \varepsilon_{22}^{0}\\ \gamma_{12}^{0} \end{array}\right]-z\left[\begin{array}{l} \frac{\partial^{2}w}{\partial x_{1}^{2}}\\ \frac{\partial^{2}w}{\partial x_{2}^{2}}\\ 2\frac{\partial^{2}w}{\partial x_{1}\partial x_{2}} \end{array}\right]+\left[\begin{array}{l} \Gamma_{1}^{\left({\bar{x}}_{3},T\right)}\frac{\partial F_{1}}{\partial x_{1}}\\ \Gamma_{2}^{\left({\bar{x}}_{3},T\right)}\frac{\partial F_{2}}{\partial x_{2}}\\ \Gamma_{1}^{\left({\bar{x}}_{3},T\right)}\frac{\partial F_{1}}{\partial x_{2}}+\Gamma_{2}^{\left({\bar{x}}_{3},T\right)}\frac{\partial F_{2}}{\partial x_{1}} \end{array}\right]$$

The first column on the right-hand side of Equation (10) shows the strain components at the mid-surface, and the following definitions apply:(11)$$\Gamma_{1}^{\left(x_{3},T\right)}=\int_{0}^{x_{3}}\frac{1}{Q_{55}^{\left({\bar{x}}_{3},T\right)}}\frac{\partial f}{\partial x_{3}}dx_{3},\;\Gamma_{2}^{\left(x_{3},T\right)}=\int_{0}^{x_{3}}\frac{1}{Q_{44}^{\left({\bar{x}}_{3},T\right)}}\frac{\partial f}{\partial x_{3}}dx_{3}$$

The stress resultants of INCCSs are determined as follows [43]:(12)$$\left(N_{ij},Q_{i},\;M_{ij}\right)=\int_{-h/2}^{h/2}\left(\sigma_{ij},\sigma_{i3},\;x_{3}\sigma_{ij}\right)dx_{3},\;(i,j=1,2)$$
where $N_{ij}$ and $Q_{i}$ are the forces, and $M_{ij}$ are the moments.

The thermal forces and moments ($N_{11}^{T},\;N_{22}^{T},\;M_{11}^{T},\;M_{22}^{T}$) are determined as follows [13,15,16,34]: (13)$$\begin{array}{l} N_{11}^{T}=\int_{-h/2}^{h/2}\left[a_{11}^{\left({\bar{x}}_{3},T\right)}\alpha_{11}^{\left({\bar{x}}_{3},T\right)}+a_{12}^{\left({\bar{x}}_{3},T\right)}\alpha_{22}^{\left({\bar{x}}_{3},T\right)}\right]\Delta Tdx_{3},\;N_{22}^{T}=\int_{-h/2}^{h/2}\left[a_{21}^{\left({\bar{x}}_{3},T\right)}\alpha_{11}^{\left({\bar{x}}_{3},T\right)}+a_{22}^{\left({\bar{x}}_{3},T\right)}\alpha_{22}^{\left({\bar{x}}_{3},T\right)}\right]\Delta Tdx_{3}\\ M_{11}^{T}=\int_{-h/2}^{h/2}\left[a_{11}^{\left({\bar{x}}_{3},T\right)}\alpha_{11}^{\left({\bar{x}}_{3},T\right)}+a_{12}^{\left({\bar{x}}_{3},T\right)}\alpha_{22}^{\left({\bar{x}}_{3},T\right)}\right]x_{3}\Delta Tdx_{3},\;M_{22}^{T}=\int_{-h/2}^{h/2}\left[a_{21}^{\left({\bar{x}}_{3},T\right)}\alpha_{11}^{\left({\bar{x}}_{3},T\right)}+a_{22}^{\left({\bar{x}}_{3},T\right)}\alpha_{22}^{\left({\bar{x}}_{3},T\right)}\right]x_{3}\Delta Tdx_{3} \end{array}$$
where $\Delta T=T-T_{0}$ is the rise in temperature against the reference temperature $T_{0}$. It should be emphasized that when $T=T_{0}$ the shell has no thermal strains.

By introducing the Airy stress function Φ as follows [43]:(14)$$N_{11}=h\frac{\partial^{2}\Phi }{\partial x_{2}^{2}},\;N_{12}=\;-h\frac{\partial^{2}\Phi }{\partial x_{1}\partial x_{2}},\;N_{22}=\;h\frac{\partial^{2}\Phi }{\partial x_{1}^{2}}$$
when the relations (6), (9), (11) and (13) are solved together, the force and moment components and the strains in the mid-surface are expressed with the functions $\Phi ,\;w,\;F_{1}$, and $F_{2}$ without taking into account the intermediate operations. Then the resulting expressions and (12) are substituted into the stability and compatibility equations for INCCSs under hydrostatic pressure in the thermal environment, and the following governing equations are derived:(15)$$\begin{array}{l} L_{11}(\Phi )+\;L_{12}(w)+L_{13}(F_{1})+L_{14}(F_{2})=0\\ L_{21}(\Phi )+\;L_{22}(w)+L_{23}(F_{1})+L_{24}(F_{2})=0\\ L_{31}(\Phi )+L_{32}(w)+L_{33}(F_{1})+L_{34}(F_{2})=0\;\\ L_{41}(\Phi )+L_{42}(w)+L_{43}(F_{2})+\;L_{44}(F_{2})=0\; \end{array}$$
where $L_{ij}(i,j=1,2,3,4)$ are described in Appendix A.

Equation (15) presents the governing equations of INCCSs under hydrostatic pressure in the thermal environment within FSDT.

## 4. Solution Procedure

Suppose that the two end edges of the cylindrical shell are simply supported; the corresponding boundary conditions are modeled as follows [13]:

At $x_{1}=0,\;L$
(16a)$$w=\frac{\partial^{2}\Phi }{\partial x_{2}^{2}}=F_{2}=M_{11}=0$$
(16b)$$\int_{0}^{2\pi R}N_{11}dx_{2}+\pi R^{2}P=0$$

Also, the closed or periodicity condition is expressed as
(17)$$\int_{0}^{2\pi R}\frac{\partial v}{\partial x_{2}}dx_{2}=0$$

The approximation functions for the above boundary conditions are searched as follows [43]:(18)$$\begin{array}{l} \Phi =C_{1}\sin(\bar{m}x_{1})\;\sin(\bar{n}x_{2}),\;w=C_{2}\sin(\bar{m}x_{1})\;\sin(\bar{n}x_{2}),\;\\ F_{1}=C_{3}\;\cos(\bar{m}x_{1})\;\sin(\bar{n}x_{2}),\;F_{2}=C_{4}\;\sin(\bar{m}x_{1})\;\cos(\bar{n}x_{2}) \end{array}$$
where $C_{i}$ denote amplitudes; $\bar{m}=\frac{m\pi }{L}$ and $\bar{n}=\frac{n}{R}$, where m and n wave numbers contained in these parameters.

Substituting (18) into the system of Equation (15), and also taking into account (1), and then using the Galerkin procedure for the resulting equations in the range $\left\{0\leq x_{1}\leq L,\;0\leq x_{2}\leq 2\pi R\right\}$, one obtains
(19)$$\left[\begin{array}{llll} A_{11} & -A_{12} & A_{13} & A_{14}\\ A_{21} & -A_{22} & A_{23} & A_{24}\\ A_{31} & -A_{32} & A_{33} & A_{34}\\ A_{41} & P\cdot A_{42} & A_{43} & A_{44} \end{array}\right]\left[\begin{array}{l} C_{1}\\ C_{2}\\ C_{3}\\ C_{4} \end{array}\right]=\left[\begin{array}{l} 0\\ 0\\ 0\\ 0 \end{array}\right]$$
where $A_{ij}(i,j=1,2,3,4)$ are parameters characterizing the INCCSs’ properties and hydrostatic pressure and are given in Appendix B.

To find the analytical expression that determines the hydrostatic buckling pressure of the INCCSs, the determinant of the square matrix of the coefficients of Equation (18), expressed by the cofactors $K_{i}(i=1,2,\ldots ,4)$, is set to zero:(20)$$A_{41}K_{1}-R{\left(0.5{\bar{m}}^{2}+{\bar{n}}^{2}\right)}^{2}K_{2}+A_{43}K_{3}+A_{44}K_{4}=0$$
where
(21)$$K_{1}=-\left|\begin{array}{lll} A_{12} & A_{13} & A_{14}\\ A_{22} & A_{23} & A_{24}\\ A_{32} & A_{33} & A_{34} \end{array}\right|,\;K_{2}=\left|\begin{array}{lll} A_{11} & A_{13} & A_{14}\\ A_{21} & A_{23} & A_{24}\\ A_{31} & A_{33} & A_{34} \end{array}\right|,\;K_{3}=-\left|\begin{array}{lll} A_{11} & A_{12} & A_{14}\\ A_{21} & A_{22} & A_{24}\\ A_{31} & A_{32} & A_{34} \end{array}\right|,\;K_{4}=\left|\begin{array}{lll} A_{11} & A_{12} & A_{13}\\ A_{21} & A_{22} & A_{23}\\ A_{31} & A_{32} & A_{33} \end{array}\right|$$

From Equation (20), one obtains
(22)$$P_{1buc}^{sdt}=\frac{A_{41}K_{1}+\;A_{43}K_{3}+A_{44}K_{4}}{E^{m}RK_{2}(0.5{\bar{m}}^{2}+{\bar{n}}^{2})}$$

Since the influence of shear stresses is ignored in the basic relations, the following expression for the hydrostatic buckling pressure of INCCSs is obtained within the CST:(23)$$\begin{array}{l} P_{1buc}^{cst}=\frac{1}{\left(0.5{\bar{m}}^{2}+{\bar{n}}^{2}\right)E^{m}R}\left\{t_{13}{\bar{m}}^{4}+\left(t_{14}+2t_{32}+t_{23}\right){\bar{m}}^{2}{\bar{n}}^{2}+t_{24}{\bar{n}}^{4}\right.+\left[{\bar{m}}^{2}/R-t_{12}{\bar{m}}^{4}\right.\\ \left.-\left(t_{11}-2t_{31}+t_{22}\right){\bar{m}}^{2}{\bar{n}}^{2}-t_{21}{\bar{n}}^{4}\right]\times \left.\frac{\left[{\bar{m}}^{2}/R+s_{23}{\bar{m}}^{4}+\left(s_{13}-s_{32}+s_{24}\right){\bar{m}}^{2}{\bar{n}}^{2}+s_{14}{\bar{n}}^{4}\right]}{s_{22}{\bar{m}}^{4}+\left(s_{12}+s_{13}+s_{21}\right){\bar{m}}^{2}{\bar{n}}^{2}+s_{11}{\bar{n}}^{4}}\right\} \end{array}$$

## 5. Numerical Examples and Discussion

The numerical values of the hydrostatic buckling pressure of INCCSs in thermal environments are presented in this section. The material used for the matrix is poly (methylmethacrylate), called PMMA, and the material used as the reinforcement element is (10, 10) armchair SWCNT with length $L^{cnt}=9.26\;\mathrm{nm}$, radius $r^{cnt}=0.68\;\mathrm{nm}$, and thickness $h^{cnt}=0.067\;\mathrm{nm}$. The material properties and efficiency parameters are evaluated as [13]
(24)$$\begin{array}{l} E^{m}=3.52-0.0034T,\;\alpha^{m}=45(1+0.0005\Delta T)\cdot 10^{-6}/K,\;\nu^{m}=0.34,\;\\ E_{11}^{cnt}=6.18387-2.86\times 10^{-3}T+4.22867\times 10^{-6}T^{2}-2.2724\times 10^{-9}T^{3}\\ E_{22}^{cnt}=7.75348-3.58\times 10^{-3}T+5.30057\times 10^{-6}T^{2}-2.84868\times 10^{-9}T^{3}\\ G_{12}^{cnt}=1.80126+0.77845\times 10^{-3}T-1.1279\times 10^{-6}T^{2}+4.93484\times 10^{-10}T^{3}\\ \alpha_{11}^{cnt}=(-1.12148+2.289\times 10^{-2}T-2.88155\times 10^{-5}T^{2}+1.13253\times 10^{-8}T^{3})\cdot 10^{-6}/K\\ \alpha_{22}^{cnt}=(5.43874-9.95498\times 10^{-4}T+3.13525\times 10^{-7}T^{2}-3.56332\times 10^{-12}T^{3})\cdot 10^{-6}/K \end{array}$$
and
(25)$$\begin{array}{l} \eta_{1}=0.137,\;\eta_{2}=1.022,\;\eta_{3}=0.715\;\mathrm{for}\;V_{cnt}^{*}=0.12;\;\eta_{1}=0.142,\;\eta_{2}=1.626,\;\eta_{3}=1.138\\ \mathrm{for}\;V_{cnt}^{*}=0.17;\;\eta_{1}=0.141,\;\eta_{2}=1.585,\;\eta_{3}=1.109\;\mathrm{for}\;V_{cnt}^{*}=0.28 \end{array}$$

It should be emphasized that in medium-length cylindrical shells (1≤L/R≤5) at R/h>100, when Formula (22) is minimized according to the wave numbers, the expression for the hydrostatic buckling pressure within FSDT gives valid results within CST. When R/h>100, the results obtained using the Formula (22) are the same as the results obtained with Formula (23) until the third digit after the decimal point. When R/h = 100, the same values are obtained up to the second digit after the decimal point. Therefore, when R/h>100, Formula (22) can also be used for the hydrostatic buckling pressure calculations within the framework of the CST. In numerical calculations, Formulas (24) and (25) are used for the material properties. Other parameters are presented in Table 1.

The accuracy of the present method for buckling analyses of cylindrical shells subjected to hydrostatic pressure is confirmed by two comparative studies. 

**Example 1.** *In this example, our results are compared with the results of Kazagi and Sridharan* *[45]* *and Shen and Noda* *[46]* *for hydrostatic buckling pressure (in psi) of cylindrical shells of different sizes made of pure metal and tabulated in Table 2. Unlike our study, Kazagi and Sridharan* *[45]* *used FEM, and Shen and Noda* *[46]* *used HSDT and the singular perturbation technique. Formula (22) is used in the calculation, taking into account that $V_{cnt}^{*}=0,\;V^{m}=1,\;E_{11}^{\left({\bar{x}}_{3},T\right)}=E_{22}^{\left({\bar{x}}_{3},T\right)}=E^{m},\;\nu_{12}=\nu_{21}=\nu^{m}$**. Some of the data used in the comparison are given in Table 2, and some of them are as follows $E^{m}=10^{7}\mathrm{psi},\;\nu^{m}=0.33,\;R=50\;h$**. The Badthorf shell parameter is presented as ${\bar{Z}}_{b}$* *in Table 2 and is defined as ${\bar{Z}}_{b}=\frac{L^{2}}{Rh}\sqrt{1-{\left(\nu^{m}\right)}^{2}}$**. The values in parentheses denote the buckling mode in Table 2. It is seen that the hydrostatic buckling pressure values are in good agreement with the results obtained by HSDT and FEM (see Refs.* *[45,46]).*

**Example 2.** *The magnitudes of the lateral buckling pressure for composite cylindrical shells patterned by CNTs with the U- and X-models based on the FSDT under two thermal environmental conditions are compared with the results of Shen* *[13]* *using HSDT and the singular perturbation technique and are presented in* Table 3*. From Formula (22), the formula for the lateral buckling pressure is obtained as follows:*$P_{buc}^{sdt}=\frac{A_{41}K_{1}+\;A_{43}K_{3}+A_{44}K_{4}}{K_{2}{\bar{n}}^{2}}$. *The material properties are taken from the study of Shen* [13] *and can be calculated from (24) and (25) at T = 300 (K). The following geometrical data are used:*
$L^{2}/Rh=100,\;h=0.002\;m,\;R=30h$
*[13].* *Since the number of longitudinal waves corresponding to the lateral buckling pressure is equal to one, it is not included in Table 3, and the values of the number of circumferential waves are shown in the table. The magnitudes of lateral buckling pressure for the U- and X-patterned nanocomposite cylinders based on the FSDT for the same circumferential wave numbers are in good agreement with the results of Ref.* *[13]* *at two thermal environmental conditions*.

Typical results of the parametric study are presented in Table 4 and Table 5 and illustrated in Figure 3, Figure 4, Figure 5 and Figure 6. In these examples, h=0.002 m and L/R=1 for moderately thick cylindrical shells under four thermal environmental conditions. 

Table 4 presents the distribution of hydrostatic buckling pressure values of the U-, V-, Λ-, and X-scheme nanocomposite cylindrical shells according to the change in $V_{cnt}^{*}$ under four thermal environmental conditions. The cylinder dimensions used are R=25h, L/R=1, h=0.002 m, and the temperature increases by 150 steps between 300 (K) and 750 (K). Table 4 shows that the increase in $V_{cnt}^{*}$ significantly increases the hydrostatic buckling pressure values in homogeneous and all inhomogeneous patterns. Although the increase in temperature from 300 (K) to 750 (K) reduces the hydrostatic buckling pressure values, the effect of the $V_{cnt}^{*}$ change remains important. For example, at $V_{cnt}^{*}$ = 0.28 and *T* = 300 (K), the influences of shear deformations on the buckling pressure in shells with U-, V-, Λ-, and X-schemes are 12.58%, 9.03%, 8.7%, and 18.43%, respectively, and at *T* = 750 (K), these effects are 24.46%, 17.28%, 18.75%, and 33.33%. When comparing $V_{cnt}^{*}$ = 0.28 and 0.17, the influences of shear deformations (SDs) for *T* = 300 (K) are less by 3.8%, 2%, 3%, and 4.3% in the U-, V-, Λ-, and X-constructs, respectively, while these the effects for *T* = 750 (K) are less approximately by 6%, 3.8%, 5.2%, and 7.8%, respectively. In all designs, the effect of SDs on hydrostatic buckling pressure values at $V_{cnt}^{*}$ = 0.12 fluctuated between the effects of $V_{cnt}^{*}$ = 0.17 and $V_{cnt}^{*}$ = 0.28. The effects of inhomogeneous patterns on the hydrostatic buckling pressure values generally increase with the rise in temperature from 300 (K) to 750 (K) in the framework of CST and are more pronounced than the influence in the FSDT. In addition, for $V_{cnt}^{*}$ = 0.17 and 0.28, the influence of the X-scheme reduces continuously with the increase in temperature in the FSDT framework, while in all other cases, those influences change irregularly. In the FSDT framework, the most pronounced effect occurs in the X-patterned cylinder with 26.43% when *T* = 300 (K) and $V_{cnt}^{*}$ = 0.28, while that effect occurs in the CST framework for the same volume fraction and patterned shell as 37.23%, at *T* = 750 (K). It has been found that as the temperature rises, its influence on the values of hydrostatic buckling pressure rises. In the same models, the increase in temperature significantly changes the values of the hydrostatic buckling pressure, and the effect of temperature is more pronounced in the frame of the FSDT. For example, when comparing the cases *T* = 300 (K) and 450 (K) in the same Λ-model, the effect of temperature on the buckling pressure values within the FSDT is 15%, 15.31%, and 14.29% for $V_{cnt}^{*}$ = 0.12, 0.17, and 0.28, respectively, while those effects decrease within CST and are 13.85%, 14.29%, and 12.32%, respectively. When comparing the cases *T* = 300 (K) and 750 (K), the effects of temperature on the buckling pressure in the same Λ-scheme are 48.33%, 47.96%, and 48.41% for $V_{cnt}^{*}$ = 0.12, 0.17, and 0.28, respectively, within the FSDT, while those influences reduce significantly and are 43.07%, 43.81%, and 42.03%, respectively, but nevertheless retain their influence within the CST. 

Table 5 and Figure 3, Figure 4, Figure 5 and Figure 6 show the distribution of hydrostatic buckling pressure values of the U-, V-, Λ-, and X-scheme nanocomposite cylindrical shells according to the change in R/h under four thermal environmental conditions. The cylinder dimensions used are $V_{cnt}^{*}=0.12,\;L/R=1,\;h=0.002\;m$; the temperature increases in steps of 150 between 300 (K) and 750 (K), and R/h increases from 20 to 40 in steps of 5. Table 5 and Figure 3, Figure 4, Figure 5 and Figure 6 show that the rise in R/h significantly reduces the hydrostatic buckling pressure values in the nanocomposite cylinders with homogeneous and all inhomogeneous patterns, whereas the corresponding circumferential wave numbers slightly increase.

Although the effect of SDs on the hydrostatic buckling pressure reduces as the R/h ratio increases, it is seen that an increase in temperature increases that effect (see Figure 3 and Figure 4). For example, the effect of SDs on the hydrostatic buckling pressure in U-, V-, Λ- and X-models in room temperature for R/h = 20 are 16.11%, 12.12%, 11.2%, and 20.65%, respectively, while those effects reduce to 5%, 5.26%, 0%, and 4.17%, respectively, at R/h = 40. For *T* = 750 (K) and R/h = 20, the effects of SDs on the hydrostatic buckling pressure in U-, V-, Λ-, and X-models are 28.74%, 22.97%, 22.22%, and 35.78%, respectively, while for R/h = 40, it is observed that those influences decreased to 9.09%, 10%, 0%, and 14.29%.

The influence of the V- and Λ-models on the hydrostatic buckling pressure decreases compared to that of the U-model, but that effect increases in the X-pattern as the R/h increases. However, an increase in temperature increases the effect of all inhomogeneous models on the hydrostatic buckling pressure. For example, at room temperature for R/h = 20, the pattern effects on the hydrostatic buckling pressure are −7.2% and −11.2% in the V- and Λ-models, respectively, while those effects reduce to −5.26% and −10.53% at R/h = 40. For *T* = 750 (K) and R/h = 20, the pattern effects on the buckling pressure in V- and Λ-models are 8.07% and −9.68%, respectively, while for R/h = 40, both pattern effects rise by −10%. For the room temperature, the effect of the X-model increases from 16.8% to 21.05% as R/h increases from 20 to 40, whereas for *T* = 750 (K), that effect rises from 12.9% to 20%.

In the FSDT framework, the influences of the same models on the hydrostatic buckling pressure rise significantly with increasing temperature. For example, for R/h = 30, the effect difference between the *T* = 300 (K) and *T* = 750 (K) temperatures on the buckling pressure load of U-, V-, Λ-, and X-modeled cylinders is around −48%. It is seen that the temperature effect is lower by around 2–12% in the framework of the CST (see Figure 5 and Figure 6).

## 6. Conclusions

The stability problem of INCCSs subjected to uniform hydrostatic pressure and under four thermal environmental conditions was investigated comparatively within the framework of two theories. The properties of the polymer-based nanocomposite forming the cylindrical shells were modeled as the function of the thickness coordinate and temperature. The effective material properties of the inhomogeneous nanocomposite cylindrical shell were determined according to the extended mixture rule. The basic relations and stability equations were derived using the generalized FSDT. By transforming the governing equations into algebraic equations with the Galerkin procedure, the expressions for the hydrostatic buckling pressure of the INCCSs were found in the framework of FSDT and CST. 

The numerical analyses have yielded the following generalizations: 

(a)The effects of inhomogeneous models on hydrostatic buckling pressure values generally increase with increasing temperature in the CST, but this effect is weakened when FSDT is used.(b)Although the increase in temperature reduces the hydrostatic buckling pressure values, the effect of the $V_{cnt}^{*}$ change remains important.(c)The effect of the X-scheme decreases continuously with the increase in temperature in the FSDT frame for $V_{cnt}^{*}$ = 0.17 and 0.28, while in all other cases, these effects change unevenly.(d)In the same models, the increase in temperature significantly changes the values of the hydrostatic buckling pressure, and the effect of temperature is more pronounced in the frame of the FSDT.(e)The rise in R/h significantly reduces the hydrostatic buckling pressure values in the nanocomposite cylinders with homogeneous and all inhomogeneous models, whereas the corresponding circumferential wave numbers slightly increase.(f)Although the effect of shear strains on the hydrostatic buckling pressure reduces as R/h increases, the increase in temperature increases that effect.(g)The influence of the V- and Λ-models on the hydrostatic buckling pressure decreases compared to that of the U-model, but that effect increases in the X-model as the R/h increases.

Analysis and comments using a closed-form solution revealed significant quantitative and qualitative changes in the stability of inhomogeneous nanocomposite cylindrical shells in the thermal environment. In order to prevent the damages that may occur in the applications and the accidents that may be caused by them, it is foreseen that the critical values revealed in the current study should be taken into account during the design phase of the structural elements. 

## Figures and Tables

**Figure 1 materials-16-04887-f001:**
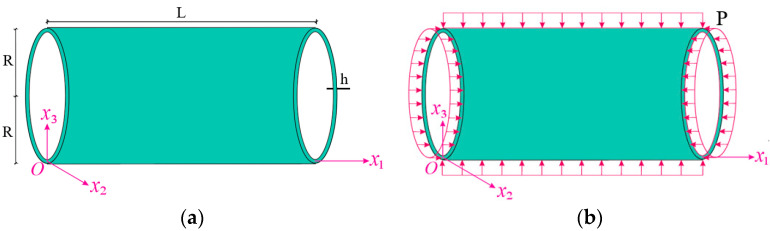
Schematic diagram of INCCS (**a**) geometry and coordinate axes and (**b**) under hydrostatic pressure.

**Figure 2 materials-16-04887-f002:**
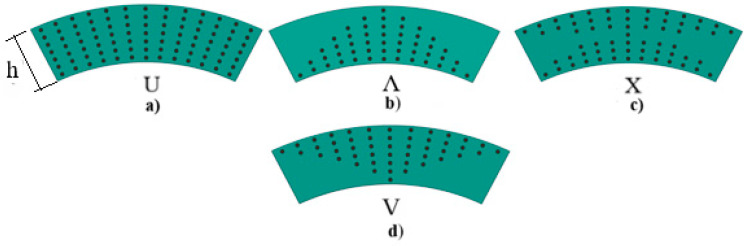
Configurations of nanocomposites: (**a**) U-, (**b**) Λ-, (**c**) X-, and (**d**) V-models.

**Figure 3 materials-16-04887-f003:**
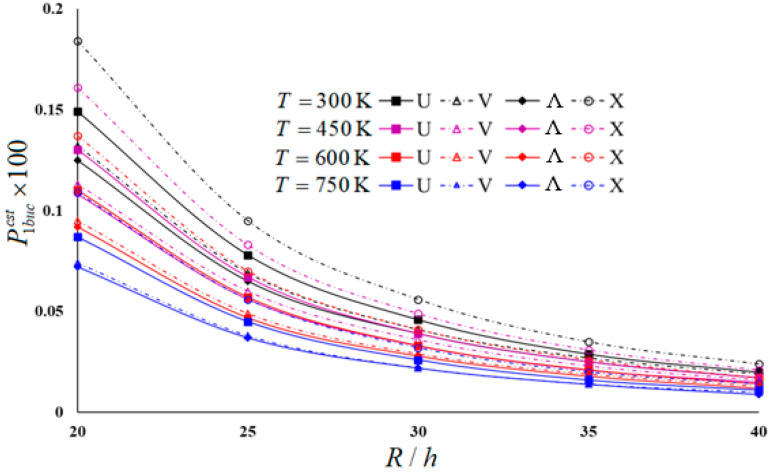
Distribution of hydrostatic buckling pressure of homogeneous and INCCSs within CST according to R/h with different temperatures.

**Figure 4 materials-16-04887-f004:**
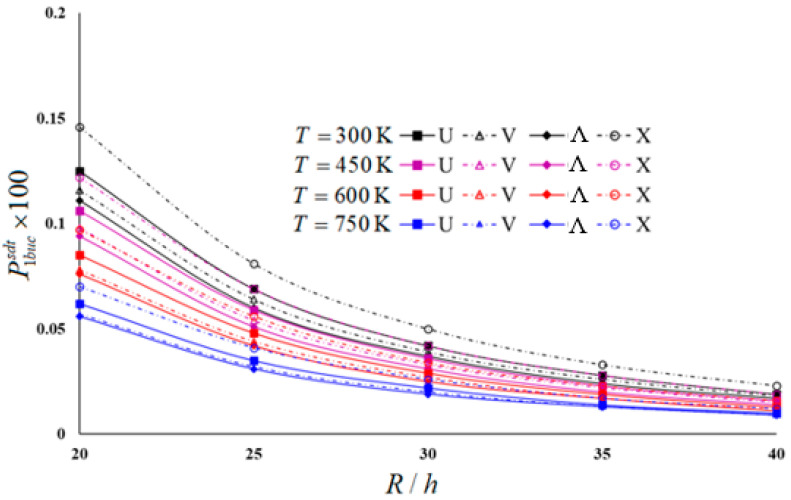
Distribution of hydrostatic buckling pressure of homogeneous and INCCSs within FSDT according to R/h with different temperatures.

**Figure 5 materials-16-04887-f005:**
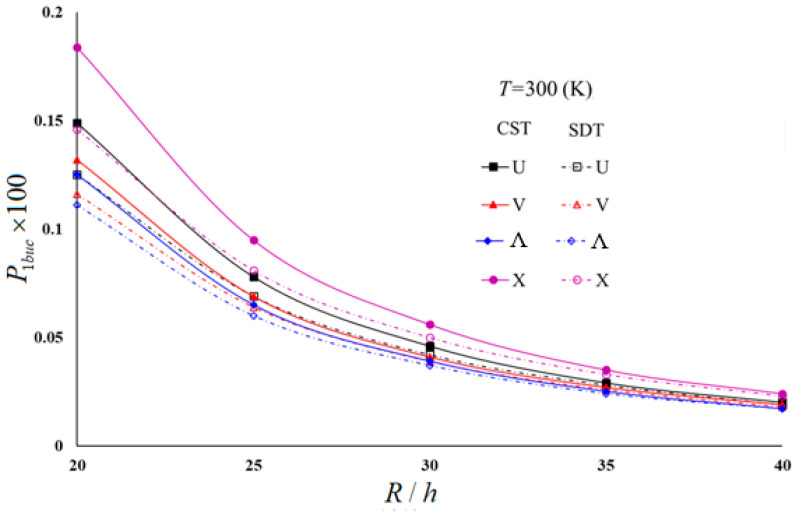
Distribution of hydrostatic buckling pressure of homogeneous and INCCSs within CST and FSDT according to R/h for *T* = 300 (K).

**Figure 6 materials-16-04887-f006:**
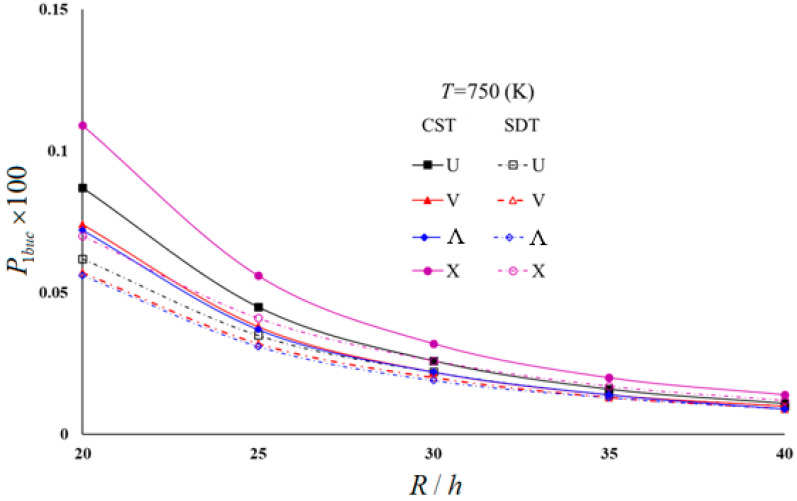
Distribution of hydrostatic buckling pressure of homogeneous and INCCSs within CST and FSDT according to R/h for *T* = 750 (K).

**Table 1 materials-16-04887-t001:** Comparison of Formulas (22) and (23) for hydrostatic buckling pressure of nanocomposite cylinders of different lengths for R/h≥100.

	$P_{1buc}^{cst}\;\times 10^{4},\;(n_{cr})$ $\mathrm{Using}\;\mathrm{Formula}\;(23);\;V_{cnt}^{*}\;=0.28$				$P_{1buc}^{sdt}\;\times 10^{4},\;(n_{cr})$ $\mathrm{Using}\;\mathrm{Formula}\;(22);\;V_{cnt}^{*}\;=0.28$			
	U	V	Λ	X	U	V	Λ	X
R/h	L/R=1, *T =* 300 (K)							
100	0.3227 (10)	0.3240 (10)	0.3040 (9)	0.4140 (10)	0.3206 (10)	0.3223 (10)	0.3030 (9)	0.4100 (10)
300	0.0195 (13)	0.0206 (13)	0.0195 (13)	0.0239 (13)	0.0195 (13)	0.0206 (13)	0.0194 (13)	0.0239 (13)
500	0.0056 (16)	0.0059 (16)	0.0057 (15)	0.00679 (15)	0.0056 (16)	0.0059 (16)	0.0057 (15)	0.00679 (15)
R/h	L/R=5, *T =* 300 (K)							
100	0.0729 (5)	0.0772 (5)	0.0754 (5)	0.0871 (5)	0.0729 (5)	0.0772 (5)	0.0754 (5)	0.0870 (5)
300	0.0051 (7)	0.0054 (7)	0.0053 (7)	0.0061 (7)	0.0051 (7)	0.0054 (7)	0.0053 (7)	0.0061 (7)
500	0.00149 (8)	0.00156 (8)	0.00155 (8)	0.00177 (8)	0.00149 (8)	0.00156 (8)	0.00155 (8)	0.00177 (8)
R/h	L/R=1, *T* = 750 (K)							
100	0.1634 (11)	0.1529 (10)	0.1450 (10)	0.2170 (11)	0.1605 (11)	0.1511 (10)	0.1440 (10)	0.2110 (11)
300	0.0085 (14)	0.0087 (14)	0.0082 (13)	0.0107 (14)	0.0085 (14)	0.0087 (14)	0.0082 (13)	0.0107 (14)
500	0.0023 (16)	0.0025 (16)	0.0023 (13)	0.0029 (16)	0.0023 (16)	0.0025 (16)	0.0023 (13)	0.0029 (16)
R/h	L/R=5, *T* = 750 (K)							
100	0.0306 (5)	0.0324 (5)	0.0313 (5)	0.0364 (4)	0.0306 (5)	0.0324 (5)	0.0313 (5)	0.0364 (4)
300	0.00221 (8)	0.00232 (8)	0.00230 (8)	0.00264 (7)	0.00221 (8)	0.00232 (8)	0.00230 (8)	0.00264 (7)
500	0.00064 (9)	0.00067 (9)	0.00067 (9)	0.00077 (9)	0.00064 (9)	0.00067 (9)	0.00067 (9)	0.00077 (9)

**Table 2 materials-16-04887-t002:** Comparison of the hydrostatic buckling pressure values with the results obtained using HSDT and FEM.

$P_{buc}^{cst}\left(in\;psi\right);\;\left(m_{cr},n_{cr}\right)$			
${\bar{Z}}_{b}$	Shen and Noda [45] HSDT	Kazagi and Sridharan [46] FEM	Present Study
50	566.09 (1,7)	560.0 (1,7)	566.02 (1,7)
100	389.62 (1,6)	385.6 (1,6)	389.60 (1,6)
500	166.77 (1,4)	165.0 (1,4)	166.77 (1,4)
1000	124.98 (1,3)	123.5 (1,3)	124.99 (1,3)
5000	56.500 (1,2)	55.90 (1,2)	56.570 (1,2)

**Table 3 materials-16-04887-t003:** Comparison the magnitudes of the lateral buckling pressure for cylindrical shells patterned by CNTs based on the FSDT with the results of Shen [13].

	$P_{buc}^{sdt}\;\left(in\;kPa\right),\;(n_{cr})$			
Pattern Type	U		X	
*T* (K)	$V_{cnt}^{*}$ = 0.12			
	Shen [13]	Present study	Shen [13]	Present study
300	474.80 (5)	473.74 (5)	558.72 (6)	557.36 (6)
500	367.35 (6)	367.29 (6)	432.75 (6)	431.33 (6)
*T* (K)	$V_{cnt}^{*}$ = 0.28			
300	943.62 6)	942.46 (6)	1234.8 (6)	1235.20 (6)
500	723.33 (6)	723.68 (6)	963.81 (6)	958.93 (6)

**Table 4 materials-16-04887-t004:** Distribution of hydrostatic buckling pressure of U-, V-, Λ- and X-scheme homogeneous and INCCSs according to $V_{cnt}^{*}$ under four thermal environmental conditions.

P1buc×100, (ncr)								
	U		V		Λ		X	
	CST	FSDT	CST	FSDT	CST	FSDT	CST	FSDT
$V_{cnt}^{*}$	*T* = 300 (K)							
0.12	0.078 (8)	0.069 (7)	0.069 (7)	0.063 (7)	0.065 (7)	0.060 (7)	0.095 (8)	0.081 (8)
0.17	0.123 (7)	0.111 (7)	0.111 (7)	0.103 (7)	0.105 (7)	0.098 (7)	0.154 (8)	0.134 (8)
0.28	0.160 (8)	0.140 (8)	0.144 (7)	0.131 (7)	0.138 (7)	0.126 (7)	0.217 (8)	0.177 (8)
$V_{cnt}^{*}$	*T* = 450 (K)							
0.12	0.067 (8)	0.059 (8)	0.060 (7)	0.054 (7)	0.056 (7)	0.051 (7)	0.083 (8)	0.069 (8)
0.17	0.107 (8)	0.095 (7)	0.095 (7)	0.087 (7)	0.090 (7)	0.083 (7)	0.134 (8)	0.113 (8)
0.28	0.140 (8)	0.119 (8)	0.124 (8)	0.111 (7)	0.121 (7)	0.108 (7)	0.191 (9)	0.149 (8)
$V_{cnt}^{*}$	*T* = 600 (K)							
0.12	0.057 (8)	0.048 (8)	0.049 (8)	0.044 (8)	0.047 (8)	0.042 (7)	0.070 (9)	0.056 (8)
0.17	0.089 (8)	0.077 (8)	0.079 (8)	0.071 (7)	0.075 (7)	0.068 (7)	0.114 (9)	0.091 (8)
0.28	0.118 (9)	0.097 (8)	0.103 (8)	0.090 (8)	0.101 (8)	0.088 (8)	0.162 (9)	0.120 (8)
$V_{cnt}^{*}$	*T* = 750 (K)							
0.12	0.045 (9)	0.035 (9)	0.038 (8)	0.032 (8)	0.037 (8)	0.031 (8)	0.056 (10)	0.041 (9)
0.17	0.070 (9)	0.057 (8)	0.061 (8)	0.052 (8)	0.059 (8)	0.051 (8)	0.090 (10)	0.067 (9)
0.28	0.094 (10)	0.071 (9)	0.081 (9)	0.067 (9)	0.080 (9)	0.065 (8)	0.129 (10)	0.086 (9)

**Table 5 materials-16-04887-t005:** Distribution of hydrostatic buckling pressure of homogeneous and INCCSs according to R/h with different temperatures.

$P_{1buc}\times 100,\;(n_{cr})$								
	U		V		Λ		X	
	CST	FSDT	CST	FSDT	CST	FSDT	CST	FSDT
R/h	*T* = 300 (K)							
20	0.149 (7)	0.125 (7)	0.132 (7)	0.116 (7)	0.125 (7)	0.111 (7)	0.184 (8)	0.146 (8)
25	0.078 (8)	0.069 (7)	0.069 (7)	0.064 (7)	0.065 (7)	0.060 (7)	0.095 (8)	0.081 (8)
30	0.046 (8)	0.042 (8)	0.041 (7)	0.039 (7)	0.039 (7)	0.037 (7)	0.056 (8)	0.050 (8)
35	0.029 (8)	0.028 (8)	0.027 (7)	0.026 (7)	0.025 (7)	0.024 (7)	0.035 (9)	0.033 (8)
40	0.020 (8)	0.019 (8)	0.019 (8)	0.018 (8)	0.017 (7)	0.017 (7)	0.024 (8)	0.023 (8)
R/h	*T* = 450 (K)							
20	0.130 (8)	0.106 (7)	0.113 (7)	0.098 (7)	0.108 (7)	0.094 (7)	0.161 (8)	0.122 (8)
25	0.067 (8)	0.059 (8)	0.060 (7)	0.054 (7)	0.056 (7)	0.051 (7)	0.083 (8)	0.069 (8)
30	0.039 (8)	0.036 (8)	0.036 (7)	0.033 (7)	0.033 (7)	0.031 (7)	0.049 (9)	0.042 (8)
35	0.025 (8)	0.023 (8)	0.023 (8)	0.022 (8)	0.021 (7)	0.020 (8)	0.031 (9)	0.028 (8)
40	0.017 (8)	0.016 (8)	0.016 (8)	0.015 (8)	0.015 (8)	0.014 (8)	0.021 (9)	0.019 (9)
R/h	*T* = 600 (K)							
20	0.110 (8)	0.085 (8)	0.095 (8)	0.078 (7)	0.092 (7)	0.076 (7)	0.137 (9)	0.097 (8)
25	0.057 (8)	0.048 (8)	0.049 (8)	0.044 (8)	0.047 (7)	0.042 (8)	0.070 (9)	0.056 (8)
30	0.033 (9)	0.029 (9)	0.029 (8)	0.027 (8)	0.028 (8)	0.025 (8)	0.041 (9)	0.034 (9)
35	0.021 (9)	0.019 (9)	0.019 (8)	0.017 (8)	0.018 (8)	0.017 (8)	0.026 (9)	0.023 (9)
40	0.014 (9)	0.013 (9)	0.013 (8)	0.012 (8)	0.012 (8)	0.011 (8)	0.017 (9)	0.016 (9)
R/h	*T* = 750 (K)							
20	0.087 (9)	0.062 (8)	0.074 (8)	0.057 (8)	0.072 (8)	0.056 (9)	0.109 (10)	0.070 (9)
25	0.045 (9)	0.035 (9)	0.038 (8)	0.032 (8)	0.037 (8)	0.031 (8)	0.056 (10)	0.041 (9)
30	0.026 (9)	0.022 (9)	0.022 (9)	0.020 (8)	0.022 (8)	0.019 (8)	0.032 (10)	0.026 (10)
35	0.016 (9)	0.014 (9)	0.014 (9)	0.013 (8)	0.014 (9)	0.013 (8)	0.020 (10)	0.017 (10)
40	0.011 (9)	0.010 (9)	0.010 (9)	0.009 (9)	0.009 (9)	0.009 (8)	0.014 (10)	0.012 (10)

## Data Availability

No data were reported in this study.

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
