# Peer review of "Stability Analysis of Shear Deformable Inhomogeneous Nanocomposite Cylindrical Shells under Hydrostatic Pressure in Thermal Environment"

_materials, 2023, doi:10.3390/ma16134887_

Round 1
Reviewer 1 Report
The manuscript presented the stability of inhomogeneous nanocomposite cylindrical shells (INCCSs) under hydrostatic pressure in a thermal environment. The analytical expressions are obtained for hydrostatic buckling pressure of INCCSs in the framework of FSDT and classical shell theory (CST) by performing a solution based on Galerkin's procedure. The manuscript contains some merits, and the content is interesting. However, some major problems must be carefully addressed and explained for reconsideration:
1. The presentation structure in the introduction to lead the research gap and discuss the new contribution is somehow unclear and still messed up. The authors are hence advised to revise further the introduction section to present deficiencies or shortcomings of other studies to make a bridge or research gap to introduce the novelty of their work instead of a long introduction paragraph.
2. The limitations of the proposed method and future research work should be mentioned in the manuscript.
3. Authors considered only uniform rise temperature environment only. Why? But in the literature, you find a lot of temperature variations.
4. Authors considered only simply supported conditions. Why? Authors should perform and compare the stability behaviour of the simply supported Nanocomposite Cylindrical Shells with clamped conditions.
5. What is the meaning of the expanded mixture rule?
6. The authors added more details about behaviour the Figures 3-6 and Table 4-5.
7. The results section should be concise and pointwise, highlighting the unique outcome, not the general one.
8. What is the potential application of the presented work? The author should highlight the application of the presented work in the abstract and conclusion.

Author Response
04.07.2023
EXPLANATION TO REVIEWER 1:
We would like to thank the highly respected Reviewer 1 for his/her improving remarks and the time spent for them.
The manuscript presented the stability of inhomogeneous nanocomposite cylindrical shells (INCCSs) under hydrostatic pressure in a thermal environment. The analytical expressions are obtained for hydrostatic buckling pressure of INCCSs in the framework of FSDT and classical shell theory (CST) by performing a solution based on Galerkin's procedure. The manuscript contains some merits, and the content is interesting. However, some major problems must be carefully addressed and explained for reconsideration:
SUGGESTION 1: The presentation structure in the introduction to lead the research gap and discuss the new contribution is somehow unclear and still messed up. The authors are hence advised to revise further the introduction section to present deficiencies or shortcomings of other studies to make a bridge or research gap to introduce the novelty of their work instead of a long introduction paragraph.
EXPLANATION 1: Thanks. In the introduction, detailed information about other studies was presented and the main difference of the article from these studies was written more clearly.
After research on nanocomposites revealed that they have extraordinary mechanical, physical and chemical properties [10-12], Shen [13] created a new concept of the stability of shells consisting of various patterned nanocomposites and opened up a new field for setting and solving the number of stability and vibration problems. This concept was experimentally confirmed by Kwon et al [14]. Following this work, Shen and co-authors [15, 16] investigated the postbuckling of nanotube-reinforced composite cylindrical shells under axial and radial mechanical loads in thermal environments using a singular perturbation technique. Brischetto and Carrera [17] proposed the classical and refined shell models for the analysis of nano-reinforced structures. Aragh [18] presented mathematical modelling of the stability of carbon nanotube-reinforced panels. Garcia-Macias et al. [19] investigated buckling analysis of functionally graded CNT-reinforced curved panels under axial compression and shear. Summary information on the studies of inhomogeneous nanocomposite structural elements, especially cylindrical shells, between 2012 and 2020 are included in review articles of the Khaniki and Ghayesh [20], Liew et al. [21] and Garg et al. [22].
In the vast majority of studies conducted on this subject in the last three years, different aspects of the stability and instability of unconstrained CNT patterned cylindrical panels and shells in different environments have been investigated by using different methods within the framework of different shell theories. Amog them, Tocci Monaco et al. [23] investgated the hygro-thermal vibrations and buckling of laminated nanoplates via nonlocal strain gradient theory. Bacciocchi [24] examined the buckling analysis of three-phase CNT/polymer/fiber functionally graded orthotropic plates and discussed the influence of the non-uniform distribution of the oriented fibers on the critical load. Hieu and Tung [25] presented the buckling of shear deformable CNT patterned cylindrical shells and toroidal shell segments under mechanical loads in thermal environments. Tocci Monaco et al. [26] reported the ritical temperatures for vibrations and buckling of magneto-electro-elastic nonlocal strain gradient plates. Cornacchia et al. [27] presented an analytical solution of linear vibrations and buckling problems of cross- and angle-ply nano plates with strain gradient theory. Tocci Monaco et al. [28] presented the trigonometric solution for the bending analysis of magneto-electro-elastic strain gradient nonlocal nanoplates in hygro-thermal environment. Izadi et al. [29] studied the torsional characteristics of CNTs by using micropolar elasticity models and molecular dynamics simulation. Hieu and Tung [30] reported the thermal buckling and postbuckling of CNT-reinforced composite cylindrical shell surrounded by an elastic medium with tangentially restrained edges. Chakraborty et al. [31] investigated the instability characteristics of damped CNT reinforced laminated shell panels subjected to in-plane excitations and thermal loading. Khayat et al. [32] investigated the effect of uncertainty sources on the dynamic instability of CNT-reinforced porous cylindrical shells integrated with piezoelectric layers under electro-mechanical loadings. Ghasemi and Soleymani [33] examined the effects of CNT distribution on the buckling of carbon nanotubes/fiber/polymer/metal hybrid laminates cylindrical shell. Avey et al. [34] presented mathematical modeling and analytical solution of thermoelastic stability problem of functionally graded nanocomposite cylinders within different theories. Shahmohammadi et al. [35] studied the nonlinear thermo-mechanical static analysis of toroidal shells made of nanocomposite/fiber reinforced composite plies surrounded by elastic medium. Sofiyev et al. [36] investigated the buckling behavior of sandwich cylindrical shells covered by functionally graded coatings with clamped boundary conditions under hydrostatic pressure. Sun et al. [37] examined post-buckling analysis of GPLs reinforced porous cylindrical shells under axial compression and hydrostatic pressure. Trang and Tung [38] investigated thermoelastic stability of thin CNT-reinforced composite cylindrical panels with elastically restrained edges under nonuniform in-plane temperature distribution. Avey et al. [39] presented the thermoelastic stability of CNT patterned conical shells under thermal loading in the framework of FSDT. Avey et al. [40] examined the mathematical modeling and solution of nonlinear vibration problem of laminated plates with CNT originating layers interacting with two-parameter elastic foundation. Izadi et al. [41] studied the bending characteristics of CNTs using micropolar elasticity models and molecular dynamics simulations. Ipek et al.[23] investigated the buckling behavior of nanocomposite plates with functionally graded properties under compressive loads in elastic and thermal environments.
The literature review reveals that the stability of INCCSs subjected to the hydrostatic pressure in thermal environment is not sufficiently investigated. In this work, the FSDT proposed by Ambartsumian [43] for the homogeneous anisotropic shells, is generalized to the INCCSs. This is one of the original aspects of the study. Unlike some studies mentioned above, the shear stress functions are used instead of shear correction factor for INCCSs in this study. It should be emphasized that in inhomogeneous nanocomposite structural elements, it is unrealistic to use the fixed-valued shear correction factor, since the value of the shear correction factor will change when the shape of the CNT patterns is changed. The Shapery model is used for the coefficients of thermal expansion, which are functions of the thickness coordinate [44]. The accuracy of the present method for buckling analyzes of cylindrical shells subjected to the hydrostatic pressure is confirmed by two comparative studies by Kazagi and Sridharan [45] using finite element method (FEM), and Shen and Noda [46] using high-order shear deformation theory (HSDT) and a singular perturbation technique.
SUGGESTION 2: The limitations of the proposed method and future research work should be mentioned in the manuscript.
EXPLANATION 2: Thanks. The proposed method will shed light on the solution of static and dynamic problems of single-layer and laminated nanocomposite structural elements in thermal environments.
SUGGESTION 3: Authors considered only uniform rise temperature environment only. Why? But in the literature, you find a lot of temperature variations.
EXPLANATION 3: Thanks. You are right. We (together and separately) conduct research on the behavior of FGM structural elements (ceramic and metal mixtures, heterogeneous isotropic materials) in linear and non-linear thermal environments. Modeling studies of nanocomposites in thermal environment is in its infancy and this study is one of the first attempts. In our opinion, other standard modifications for nanocomposites can be considered after certain steps.
SUGGESTION 4: Authors considered only simply supported conditions. Why? Authors should perform and compare the stability behaviour of the simply supported Nanocomposite Cylindrical Shells with clamped conditions.
EXPLANATION 4: Thanks. It's an interesting question. Analytical solution of stability and vibration problems of clamped nanocomposite shells within the framework of shear deformation theory is very difficult. In clamped boundary conditions, the determination of approximation functions is particularly complicated. In some previous studies, problems were solved by considering the clamped boundary conditions in nanocomposite shells, but all of these problems were solved by numerical methods (Shen [13, 15,16] etc.). It should be emphasized that there are two studies that we did in 2023 on the stability and vibration of cylindrical shells in clamped boundary condition without considering the thermal environment effect. Therefore, in numerical solutions, stability and vibration problems of structural elements under fixed and simply supported boundary conditions can be done in an article. In analytical solutions to this problem, it is more convenient to do them separately, because all operations must be done separately. In addition, in thermal environments, it may be difficult for built-in boundary conditions to maintain their rigidity.
SUGGESTION 5: What is the meaning of the expanded mixture rule?
EXPLANATION 5: Thanks. This is a typo. This error has been corrected in the revised paper.
SUGGESTION 6: The authors added more details about behaviour the Figures 3-6 and Table 4-5.
EXPLANATION 6: Thanks. Some improvements and comments were made in this section.
SUGGESTION 7: The results section should be concise and pointwise, highlighting the unique outcome, not the general one.
EXPLANATION 7: Thanks. The following interpretations are added to the conclusion of the revised manuscript.
The numerical analyzes have yielded the following generalizations:
- The effects of inhomogeneous models on hydrostatic buckling pressure values generally increase with increasing temperature in the CST, but this effect is weakened when FSDT is used.
- Although the increase of temperature reduces the hydrostatic buckling pressure values, the effect of the change remains important.
- The effect of the X-scheme decreases continuously with the increase in temperature in the FSDT frame for =0.17 and 0.28, while in all other cases these effects change unevenly.
- In the same models, the increase of temperature significantly changes the values of the hydrostatic buckling pressure, and the effect of temperature is more pronounced in the frame of the FSDT.
- The rises of significantly reduces the hydrostatic buckling pressure values in the nanocomposite cylinders with homogeneous and all inhomogeneous models, whereas the corresponding circumferential wave numbers slightly increment.
- Although the effect of shear strains on the hydrostatic buckling pressure reduces as the increment, the increment temperature rises that effect.
- The influence of the V- and Λ-models on the hydrostatic buckling pressure decreases compared to the U-model, but that effect increases in the X-model, as the
Analysis and comments using a closed-form solution reveal significant quantitative and qualitative changes in the stability of inhomogeneous nanocomposite cylindrical shells in the thermal environment. In order to prevent the damages that may occur in the applications and the accidents that may be caused by them, it is foreseen that the critical values revealed in the current study should be taken into account during the design phase of the structural elements.
SUGGESTION 8: What is the potential application of the presented work? The author should highlight the application of the presented work in the abstract and conclusion.
EXPLANATION 8: All of these suggestions were considered and developed in different parts of the revised article.
Also, the following comments arere added to the conclusions.
Analysis and comments using a closed-form solution reveal significant quantitative and qualitative changes in the stability of inhomogeneous nanocomposite cylindrical shells in the thermal environment. In order to prevent the damages that may occur in the applications and the accidents that may be caused by them, it is foreseen that the critical values revealed in the current study should be taken into account during the design phase of the structural elements.

Reviewer 2 Report
This work concerns the stability analysis of cylindrical shells made of carbon nanotubes (CNT). The Authors solve this problem analytically with the use of Airy stress functions and harmonic expansion; their solutions satisfactorily agree with some other models available in the literature. This problem gained recently remarkable importance because of the wide applications of CNTs in engineering applications and has been presented in many papers elsewhere.
Although this manuscript generally deserves publication in this journal, the Authors need to discuss in detail a few issues:
[1] Shapery theory has been developed more than 50 years ago when nanotubes were unknown so its application in this case should be commented in detail - application of theory relevant to traditional composite materials in the area of nanocomposites may not be so straightforward;
[2] Modified Donnell theory has been mentioned in the conclusions but no previous explanations have been provided - please postpone this phrase or expand in the mathematical model main assumptions of this theory;
[3] an overview of the literature is provided in Example 1 - please move these sentences to the Introduction section;
[4] The Badthorf shell parameter appears for the first time in Table 1, so the possible repeating of this model by the Authors starts to be difficult.
This work needs thorough reading and modification of some grammar and editing errors.
Author Response
EXPLANATION TO REVIEWER 1:
We would like to thank the highly respected Reviewer 1 for his/her improving remarks and the time spent for them.
Comments and Suggestions for Authors
This work concerns the stability analysis of cylindrical shells made of carbon nanotubes (CNT). The Authors solve this problem analytically with the use of Airy stress functions and harmonic expansion; their solutions satisfactorily agree with some other models available in the literature. This problem gained recently remarkable importance because of the wide applications of CNTs in engineering applications and has been presented in many papers elsewhere.
Although this manuscript generally deserves publication in this journal, the Authors need to discuss in detail a few issues:
SUGGESTION [1]: Shapery theory has been developed more than 50 years ago when nanotubes were unknown so its application in this case should be commented in detail - application of theory relevant to traditional composite materials in the area of nanocomposites may not be so straightforward;
EXPLANATION 1: Thanks. You are right.
Nanocomposites belong to the class of orthotropic materials. Therefore, the Shapery model for thermal expansion coefficients is valid for the traditional and new generation composite orthotropic materials. The Shapery model was adapted to the nanocomposites by Shen (2009). Although the adaptation of thermal expansion coefficients according to the Shapery model has not been emphasized in all studies since 2010 on the structural elements in the thermal environment composed of nanocomposite orthotropic or anzitropic materials, they are based on the Shapery model. Just as the stress-strain relations of nanocomposite structural elements are subject to the generalized Hooke rule, the thermal expansion coefficient is an extension of the Shapery model to nanocomposites. Also, I agree with your opinion that the theory about traditional composite materials in the field of nanocomposites is not so easy to apply, but for thermal expansion coefficients, this model has found its place for orthotropic heterogeneous nanocomposites as follows:
(4)
Shen H-S. Nonlinear bending of functionally graded carbon nanotubereinforced
composite plates in thermal environments. Compos Struct2009;91:9–19.
SUGGESTION [2]: Modified Donnell theory has been mentioned in the conclusions but no previous explanations have been provided - please postpone this phrase or expand in the mathematical model main assumptions of this theory;
EXPLANATION [2]: Thanks. It is more logical to remove “the Modified Donnell theory” statement from the conclusion.
SUGGESTION [3]: an overview of the literature is provided in Example 1 - please move these sentences to the Introduction section;
EXPLANATION [3]: Thanks. The following sentences are added to an introduction in the revised manuscript:
“The Shapery model is used for the coefficients of thermal expansion, which are functions of the thickness coordinate [44]. The accuracy of the present method for buckling analyzes of cylindrical shells subjected to the hydrostatic pressure is confirmed by two comparative studies by Kazagi and Sridharan [45] using finite element method (FEM), and Shen and Noda [46] using high-order shear deformation theory (HSDT) and a singular perturbation technique.”
SUGGESTION [4]: The Badthorf shell parameter appears for the first time in Table 1, so the possible repeating of this model by the Authors starts to be difficult.
EXPLANATION [4]: Thanks. You are right. A minor change was made in Table 2. It must be “The following geometrical data are used: [13]” instead of “The following geometrical data are used: [13]”.
Comments on the Quality of English Language
This work needs thorough reading and modification of some grammar and editing errors.
Thanks; we have improved the readability of the manuscript and checked for any typos.

Reviewer 3 Report
This paper focuses on the stability of inhomogeneous nanocomposite cylindrical shells under hydrostatic pressure in thermal environment. This topic is timely and important considering the increasing application of shell structures which are exposed to pressure and temperature.
The structure of this paper is appropriate. References are the most recent and relevant in the field.
In the 'Numerical examples and Discussion' section, even though the results have been well presented there is no extended discussion as to what physical mechanism lead to the observed results, what are its implications and new findings does the current research portray since most part of the discussion relates to an already establish phenomenon or results.
In Abstract section methods are properly described. However, the article's main findings and main interpretations were not presented.
The 'Introduction' section: "[14-19]", "[23-42]". In the Introduction, literature review, each citation should be done individually for a single reference, clubbing of more than one referred articles by one single statement for citation as it is done in several cases should be avoided otherwise it would be inferred that citations are done only for the formality without having focused and precise relevance. Break these sentences into parts or individual sentences. For example, ... [...], ... [...], etc. Or one reference - one sentence.
Line 83. The sentence "The inhomogeneous nanocomposite cylindrical shell subjected to hydrostatic pressure: ... " is incomprehensible. There are many places in the manuscript that need linguistic correction.
line 91: ‘We assume ...’ A scientific article suggests using the passive voice: It is assumed ... or It was assumed ... Please check the whole manuscript.
Please add reference(s) in the sentence: Since the effective Poisson ratio and density are weakly dependent on temperature change and location, they are considered constant.
line 94: Poisson ratio -> Poisson's ratio (in the entire manuscript).
What is the difference between Vcmt and Vcnt (Eq. 1)?
line 103: The "CNT" for Young's modulus should be lowercase.
line 109: "MD". All acronyms must be defined when first used.
In Governing Equations section the authors say :The stress-strain relationships of inhomogeneous nanocomposite cylindrical shells in the thermal environments within FSDT can be expressed as [13, 24, 34, 39]". However, the reviewer did not find this equation in [13] and [24]. All references should be selected so that they correspond to the presented content.
As the authors write, the evaluated values of parameters (Tables 2 and 3) were in good agreement with the results from references. If so, what is the novelty of this manuscript compared to previously developed methods?
The ‘Conclusions’ section: It is suggested to add quantitative conclusions.
There are many places in the manuscript that need linguistic correction.
Author Response
04.07.2023
EXPLANATION TO REVIEWER 2:
We would like to thank the highly respected Reviewer 2 for his/her improving remarks and the time spent for them.
This paper focuses on the stability of inhomogeneous nanocomposite cylindrical shells under hydrostatic pressure in thermal environment. This topic is timely and important considering the increasing application of shell structures which are exposed to pressure and temperature.
The structure of this paper is appropriate. References are the most recent and relevant in the field.
In the 'Numerical examples and Discussion' section, even though the results have been well presented there is no extended discussion as to what physical mechanism lead to the observed results, what are its implications and new findings does the current research portray since most part of the discussion relates to an already establish phenomenon or results.
In Abstract section methods are properly described. However, the article's main findings and main interpretations were not presented.
EXPLANATION: Thanks. There are three important aspects of this study, each of which is included in the abstract:
a) The first-order shear deformation theory (FSDT) for the homogeneous orthotropic shell is generalized to the inhomogeneous shell theory.
b)Analytical expressions for hydrostatic buckling pressure of INCCSs are obtained for the first time in the framework of the FSDT.
c) All analyzes and comments are original.
SUGGESTION 1: The 'Introduction' section: "[14-19]", "[23-42]". In the Introduction, literature review, each citation should be done individually for a single reference, clubbing of more than one referred articles by one single statement for citation as it is done in several cases should be avoided otherwise it would be inferred that citations are done only for the formality without having focused and precise relevance. Break these sentences into parts or individual sentences. For example, ... [...], ... [...], etc. Or one reference - one sentence.
EXPLANATION 1: Thanks. The necessary corrections have been made in the introduction part and are presented below:
After research on nanocomposites revealed that they have extraordinary mechanical, physical and chemical properties [10-12], Shen [13] created a new concept of the stability of shells consisting of various patterned nanocomposites and opened up a new field for setting and solving the number of stability and
vibration problems. This concept was experimentally confirmed by Kwon et al [14]. Following this work, Shen and co-authors [15, 16] investigated the postbuckling of nanotube-reinforced composite cylindrical shells under axial and radial mechanical loads in thermal environments using a singular perturbation technique. Brischetto and Carrera [17] proposed the classical and refined shell models for the analysis of nano-reinforced structures. Aragh [18] presented mathematical modelling of the stability of carbon nanotube-reinforced panels. Garcia-Macias et al. [19] investigated buckling analysis of functionally graded CNT-reinforced curved panels under axial compression and shear. Summary information on the studies of inhomogeneous nanocomposite structural elements, especially cylindrical shells, between 2012 and 2020 are included in review articles of the Khaniki and Ghayesh [20], Liew et al. [21] and Garg et al. [22].
In the vast majority of studies conducted on this subject in the last three years, different aspects of the stability and instability of unconstrained CNT patterned cylindrical panels and shells in different environments have been investigated by using different methods within the framework of different shell theories. Amog them, Tocci Monaco et al. [23] investgated the hygro-thermal vibrations and buckling of laminated nanoplates via nonlocal strain gradient theory. Bacciocchi [24] examined the buckling analysis of three-phase CNT/polymer/fiber functionally graded orthotropic plates and discussed the influence of the non-uniform distribution of the oriented fibers on the critical load. Hieu and Tung [25] presented the buckling of shear deformable CNT patterned cylindrical shells and toroidal shell segments under mechanical loads in thermal environments. Tocci Monaco et al. [26] reported the ritical temperatures for vibrations and buckling of magneto-electro-elastic nonlocal strain gradient plates. Cornacchia et al. [27] presented an analytical solution of linear vibrations and buckling problems of cross- and angle-ply nano plates with strain gradient theory. Tocci Monaco et al. [28] presented the trigonometric solution for the bending analysis of magneto-electro-elastic strain gradient nonlocal nanoplates in hygro-thermal environment. Izadi et al. [29] studied the torsional characteristics of CNTs by using micropolar elasticity models and molecular dynamics simulation. Hieu and Tung [30] reported the thermal buckling and postbuckling of CNT-reinforced composite cylindrical shell surrounded by an elastic medium with tangentially restrained edges. Chakraborty et al. [31] investigated the instability characteristics of damped CNT reinforced laminated shell panels subjected to in-plane excitations and thermal loading. Khayat et al. [32] investigated the effect of uncertainty sources on the dynamic instability of CNT-reinforced porous cylindrical shells integrated with piezoelectric layers under electro-mechanical loadings. Ghasemi and Soleymani [33] examined the effects of CNT distribution on the buckling of carbon nanotubes/fiber/polymer/metal hybrid laminates cylindrical shell. Avey et al. [34] presented mathematical modeling and analytical solution of thermoelastic stability problem of functionally graded nanocomposite cylinders within different
theories. Shahmohammadi et al. [35] studied the nonlinear thermo-mechanical static analysis of toroidal shells made of nanocomposite/fiber reinforced composite plies surrounded by elastic medium. Sofiyev et al. [36] investigated the buckling behavior of sandwich cylindrical shells covered by functionally graded coatings with clamped boundary conditions under hydrostatic pressure. Sun et al. [37] examined post-buckling analysis of GPLs reinforced porous cylindrical shells under axial compression and hydrostatic pressure. Trang and Tung [38] investigated thermoelastic stability of thin CNT-reinforced composite cylindrical panels with elastically restrained edges under nonuniform in-plane temperature distribution. Avey et al. [39] presented the thermoelastic stability of CNT patterned conical shells under thermal loading in the framework of FSDT. Avey et al. [40] examined the mathematical modeling and solution of nonlinear vibration problem of laminated plates with CNT originating layers interacting with two-parameter elastic foundation. Izadi et al. [41] studied the bending characteristics of CNTs using micropolar elasticity models and molecular dynamics simulations. Ipek et al.[23] investigated the buckling behavior of nanocomposite plates with functionally graded properties under compressive loads in elastic and thermal environments.
SUGGESTION 2: Line 83. The sentence "The inhomogeneous nanocomposite cylindrical shell subjected to hydrostatic pressure: ... " is incomprehensible. There are many places in the manuscript that need linguistic correction.
EXPLANATION 2: Thanks. You are right. The sentence has been modified as follows: “The hydrostatic pressure acting on the inhomogeneous nanocomposite cylindrical shell is expressed as:”
SUGGESTION 3:line 91: ‘We assume ...’ A scientific article suggests using the passive voice: It is assumed ... or It was assumed ... Please check the whole manuscript.
EXPLANATION 3: Thanks. You are right. The sentence has been modified in revised paper as follows: “It is assumed that the material properties of the CNTs and the matrix are temperature dependent.”
SUGGESTION 4: Please add reference(s) in the sentence: Since the effective Poisson ratio and density are weakly dependent on temperature change and location, they are considered constant.
EXPLANATION 4: The link has been added and the sentence has been modified in revised paper as follows: “Since the effective Poisson’s ratio and density are weakly dependent on temperature change and location, they are considered constant [13].”
SUGGESTION 5: line 94: Poisson ratio -> Poisson's ratio (in the entire manuscript).
EXPLANATION 5: "Poisson's ratio" is written instead of "Poisson ratio" in revised paper.
SUGGESTION 6: What is the difference between Vcmt and Vcnt (Eq. 1)?
EXPLANATION 6: Thanks. You are right. Misspelled, Vcnt written instead of Vcmt in revised paper.
SUGGESTION 7: line 103: The "CNT" for Young's modulus should be lowercase.
EXPLANATION 7: Thanks. CNT is written in lowercase in revised paper.
SUGGESTION 8: line 109: "MD". All acronyms must be defined when first used.
EXPLANATION 8: Thanks. Molecular dynamics (MD) is written instead of MD in revised paper.
SUGGESTION 9: In Governing Equations section the authors say :The stress-strain relationships of inhomogeneous nanocomposite cylindrical shells in the thermal environments within FSDT can be expressed as [13, 24, 34, 39]". However, the reviewer did not find this equation in [13] and [24]. All references should be selected so that they correspond to the presented content.
EXPLANATION 9: Thanks. You are right. In the revised manuscript, the [13] and [24] have been removed. It should be emphasized that the stress-strain relations for nanocomposite shells are used in those articles also, although it does not seem obvious. Otherwise the authors would not be able to derive the basic equations.
SUGGESTION 10: As the authors write, the evaluated values of parameters (Tables 2 and 3) were in good agreement with the results from references. If so, what is the novelty of this manuscript compared to previously developed methods?
EXPLANATION 10: Thanks. We have shown this difference in the interpretations of Table 1 that there are different theories or solution methods. We have highlighted this point in Table 2. The following notes are added to the introduction and the comparison part of the revised manuscript.
The accuracy of the present method for buckling analyzes of cylindrical shells subjected to the hydrostatic pressure is confirmed by two comparative studies by Kazagi and Sridharan [45] using finite element method (FEM), and Shen and Noda [46] using high-order shear deformation theory (HSDT) and a singular perturbation technique.
SUGGESTION 11: The ‘Conclusions’ section: It is suggested to add quantitative
conclusions.
EXPLANATION 11: Thanks. You are rigjt. The following interpretations are
added to the conclusion of the revised manuscript.
The numerical analyzes have yielded the following generalizations:
a) The effects of inhomogeneous models on hydrostatic buckling pressure values
generally increase with increasing temperature in the CST, but this effect is
weakened when FSDT is used.
b) Although the increase of temperature reduces the hydrostatic buckling pressure
values, the effect of the *
cnt V change remains important.
c) The effect of the X-scheme decreases continuously with the increase in
temperature in the FSDT frame for *
cnt V =0.17 and 0.28, while in all other cases
these effects change unevenly.
d) In the same models, the increase of temperature significantly changes the
values of the hydrostatic buckling pressure, and the effect of temperature is
more pronounced in the frame of the FSDT.
e) The rises of /Rh significantly reduces the hydrostatic buckling pressure values
in the nanocomposite cylinders with homogeneous and all inhomogeneous
models, whereas the corresponding circumferential wave numbers slightly
increment.
f) Although the effect of shear strains on the hydrostatic buckling pressure
reduces as the / Rh increment, the increment temperature rises that effect.
g) The influence of the V- and Λ-models on the hydrostatic buckling pressure
decreases compared to the U-model, but that effect increases in the X-model,
as the / Rh increases.
SUGGESTION 12: Comments on the Quality of English Language. There are
many places in the manuscript that need linguistic correction.
EXPLANATION 12: Thanks. We have improved the readability of the
manuscript and checked for any typos.

Reviewer 4 Report
1. Check the name of the axes in Figure 1. Show L, R and h in Figure 1.
2. Explain how formula (1) is obtained.
3. Check the statement in line 94. In fact, the Poisson ratio of matrix varies with temperature. For example, at a temperature close to the glass transition temperature, the Poisson ratio is about 0.5.
4. Decipher the abbreviations: MD, SDT
5. It is recommended to give a reference to the literary source in line 110.
6. In section 5. Numerical examples and Discussion, check the temperatures used. You must specify T0. It should be taken into account that the melting point of PMMA is about 430 K. Therefore, calculations above this temperature have no practical significance.
7. It is necessary to discuss the practical significance of the results obtained. For what types of products can the obtained results be applied? What technologies are possible to produce such products?
8. How do you plan to continue this research? For example, conducting an experimental validation of the proposed model.
Author Response
04.07.2023
EXPLANATION TO REVIEWER 3:
We would like to thank the highly respected Reviewer 3 for his/her improving remarks and the time spent for them.
SUGGESTION 1: Check the name of the axes in Figure 1. Show L, R and h in Figure 1.
EXPLANATION 1: In Figure 1, the symbols of the axes were checked and typos are corrected. In Figure 1, the length, radius and thickness of the cylindrical shell are shown as L, R and h, respectively.
The notes on the geometry of the inhomogeneous nanocomposite cylindrical shell subjected to the hydrostatic pressure is illustrated in Figure 1. The geometrical parameters such as length, radius and thickness of the INCCS are designated by , and , respectively (Figure 1a). The shell displacements surface in the and directions are designated by and , respectively.
- a) b)
Figure 1. Schematic diagram of INCCS a) geometry and coordinate axes, and b) under a hydrostatic pressure
The hydrostatic pressure acting on the inhomogeneous nanocomposite cylindrical shell is expressed as (Figure 1b) [47]:
(1)
.
SUGGESTION 2: Explain how formula (1) is obtained.
EXPLANATION 2: The formula (1) is found in fundamental books and its derivation is clearly shown in the study of the Volmir [47]. This link has been added to the revised manuscript.
SUGGESTION 3: Check the statement in line 94. In fact, the Poisson ratio of matrix varies with temperature. For example, at a temperature close to the glass transition temperature, the Poisson ratio is about 0.5.
EXPLANATION 3: Since Poisson's ratio is expressed as , it is not possible to change it theoretically. Because does not depend on the temperature. This fact is also used in all references in the references section of our article. It should be emphasized that and are different concepts. is provided only in uniform distribution.
SUGGESTION 4: Decipher the abbreviations: MD, SDT
EXPLANATION 4: Thanks. In the revised manuscript, MD was explained as molecular dynamics (MD). In the revised manuscript, shear deformation theory (SDT) was replaced by first order shear deformation theory (FSDT).
SUGGESTION 5: It is recommended to give a reference to the literary source in line 110.
EXPLANATION 5: The following reference has been added to the end of the sentence in the revised manuscript: “The magnitudes of the CNT efficiency parameter are determined by comparing the modulus of elasticity of the nanocomposites obtained from the molecular dynamics simulation with those estimated from the mixing rule [13].”
SUGGESTION 6: In section 5. Numerical examples and Discussion, check the temperatures used. You must specify T0. It should be taken into account that the melting point of PMMA is about 430 K. Therefore, calculations above this temperature have no practical significance.
EXPLANATION 6: Thanks. I respect your opinion, but I don't agree with all of it. Many studies support our study [13-16].
SUGGESTION 7: It is necessary to discuss the practical significance of the results obtained. For what types of products can the obtained results be applied? What technologies are possible to produce such products?
EXPLANATION 7: Thanks. The extraordinary mechanical properties [1,2], design and production technology [3-5], advantages and application areas of these products [6-9] are clearly defined in the following sources in the references and information about these issues is given in the introduction. In this study, we examined the stability behavior of cylindrical shells made of CNT patterned composites under hydrostatic pressure load by performing mathematical modeling during the design and made suggestions.
- …………………..
- Arash, B., Wang, Q., Varadan, V.K. Mechanical properties of carbon nanotube/polymer composites. Scientific Rep. 2014, 4, 6479.
- Dubey, K.A., Hassan, P.A., Bhardwaj, Y.K. High performance polymer nanocomposites for structural applications. Mater. Extreme Condit. Elsevier, 2017, 159-194.
- Akpan, E.I., Shen, X., Wetzel, B., Friedrich, K. Design and synthesis of polymer nanocomposites. In book: Polymer Compos. Functional. Nanopartic., Elsevier 2019, 47-83.
- Müller, K, Bugnicourt, E., Latorre, M., Jorda, M., Echegoyen Sanz, Y., Lagaron, J.M., Miesbauer, O., Bianchin, A., Hankin, S., Bölz, U., Pérez, G., Jesdinszki, M., Lindner, M., Scheuerer, Z., Castelló, S., Schmid, M. Review on the processing and properties of polymer nanocomposites and nanocoatings and their applications in the packaging, automotive and solar energy fields. Nanomaterials2017, 7, 74.
- Chandra, A.K., Kumar, R. Polymer nanocomposites for automobile engineering applications. In book: Propert. Appl. Polymer Nanocompos. Springer 2017, 139-172.
- Iqbal, A., Saeed, A., Ul-Hamid, A. A review featuring the fundamentals and advancements of polymer/CNT nanocomposite application in aerospace industry. Polymer Bulletin 2021, 78 (1), 539-557.
- Romero, F.D., Bustamante, T.M, Plasencia, F.B., Lozano, A.E., Bucio, E. Recent trends in magnetic polymer nanocomposites for aerospace applications: A Review. Polymers 2022, 14 (19), 4084.
- Shah, V., Bhaliya, J., Patel, G.M., Deshmukh, K. Advances in polymeric nanocomposites for automotive applications: A review. Adv. Technol. 2022, 33(10), 3023‐3048.
.
SUGGESTION 8: How do you plan to continue this research? For example, conducting an experimental validation of the proposed model.
EXPLANATION 8: Thanks. Our results generally use research constructors engaged in research and application in space and aircraft research centers. Our task is to help prevent previously undetected accidents and prevent serious financial losses by examining the behavioral performance of materials in different environments by simulating during design by doing mathematical modeling.
Thank you for your constructive review.

Round 2
Reviewer 1 Report
Authors have incorporated all the suggestions. Manuscript can be accepted for publication.